# Bioactive Compounds Produced by Endophytic Microorganisms Associated with Bryophytes—The “Bryendophytes”

**DOI:** 10.3390/molecules28073246

**Published:** 2023-04-05

**Authors:** Mateusz Stelmasiewicz, Łukasz Świątek, Simon Gibbons, Agnieszka Ludwiczuk

**Affiliations:** 1Department of Pharmacognosy with the Medicinal Plant Garden, Medical University of Lublin, 20-093 Lublin, Poland; aludwiczuk@pharmacognosy.org; 2Department of Virology with Viral Diagnostics Laboratory, Medical University of Lublin, 20-093 Lublin, Poland; 3Centre for Natural Products Discovery (CNPD), Liverpool John Moores University, Liverpool L3 3AF, UK; s.gibbons@ljmu.ac.uk

**Keywords:** bryophytes, endophytic microorganisms, bryendophytes, symbiosis, biomolecules

## Abstract

The mutualistic coexistence between the host and endophyte is diverse and complex, including host growth regulation, the exchange of substances like nutrients or biostimulants, and protection from microbial or herbivore attack. The latter is commonly associated with the production by endophytes of bioactive natural products, which also possess multiple activities, including antibacterial, insecticidal, antioxidant, antitumor, and antidiabetic properties, making them interesting and valuable model substances for future development into drugs. The endophytes of higher plants have been extensively studied, but there is a dearth of information on the biodiversity of endophytic microorganisms associated with bryophytes and, more importantly, their bioactive metabolites. For the first time, we name bryophyte endophytes “bryendophytes” to elaborate on this important and productive source of biota. In this review, we summarize the current knowledge on the diversity of compounds produced by endophytes, emphasizing bioactive molecules from bryendophytes. Moreover, the isolation methods and biodiversity of bryendophytes from mosses, liverworts, and hornworts are described.

## 1. Introduction

Bryophytes are the second most diverse group of plants after the flowering plants and are considered to be the oldest terrestrial plants [1]. As the first inhabitants of terrestrial habitats, they were frequently exposed to adverse environmental conditions, such as pathogen attacks and insect predation, among others [2]. In general, bryophytes display a high degree of chemical diversification [3,4], suggesting that natural products may play an important role in bryophyte-environment interactions [2,5]. It should also be mentioned that endophytes associated with bryophyte species are able to synthesize bioactive compounds and consequently contribute, in part, to the control of microbial or herbivore attack [6].

Endophytic microorganisms promote the growth of host plants through the direct production of secondary metabolites, which increase the resistance of plants to biotic and abiotic stresses. In addition, they are able to biosynthesize medically important components that were initially thought to be produced only by the host plant [7]. Most of the research related to plant endophytes has focused on higher plants, and their diversity in lower plants such as the bryophytes has to date been neglected [8]. However, a small yet diverse series of studies has shown that an abundance of endophytic bacterial and fungal communities in bryophytes have varied roles in host physiology, pharmaceuticals, ecology, and agriculture [9,10,11].

This review highlights the importance of studying endophytic microorganisms associated with plants, especially bryophytes. For the first time, we describe this fascinating class of microbes as bryendophytes to capture the essence of their ecological association. Particularly, we focus on their isolation, cultivation, and the chemical diversity of their bioactive compounds by accessing the pertinent information through the PubMed, Scopus, Web of Science, and Google Scholar databases.

Endophytes represent a ubiquitous world in plants. They have established a mutually beneficial relationship with host plants during long-term coevolution. The endophytic community encompasses a wide variety of microbial species, constituting a complex microecosystem. Several reports have shown that endophytes enhance the fitness of their host plants through the direct production of bioactive metabolites, which are involved in protecting the host against herbivores and pathogenic microbes. Furthermore, endophytes are also able to biosynthesize medicinally important natural products [12,13]. These secondary metabolites, isolated from plant endophytes, have attracted increasing attention because they can serve as antibiotics, insecticidal agents, natural antioxidants, antitumor agents, and antidiabetic products, to name just a few therapeutic areas [9,14]. In the following sections, the procedures for endophyte isolation and an overview of the characteristics of the compounds from these microorganisms are described.

## 2. Isolation of Endophytes

The rhizosphere, a term introduced in 1904 by Lorenz Hiltner, an agronomist and plant physiologist, describes the interface between the plant root system and the soil inhabited by a versatile microbial community. Presently, this term has evolved, and the rhizosphere has been divided into three neighboring zones: the endo-rhizosphere, where microorganisms occupy free space between plant cells; the rhizoplane, localized adjacent to the root epidermal cells; and the ecto-rhizosphere, which surrounds the rhizoplane and extends into the surrounding soil, colonized by free-living or non-symbiotic microorganisms [15]. Since a diverse microbial population occupies the rhizosphere, the isolation of endophytes, which are located in the endo-rhizosphere, must include appropriate measures to eliminate possible microbial contamination originating from the rhizoplane and ecto-rhizosphere.

The procedures for endophyte isolation described in the literature vary; hence, different approaches will be evaluated in this review. The collected plant material should be thoroughly washed in running water to remove possible dust or soil contaminants and carefully examined for any visible injuries. At this point, any injured plant specimens should be discarded [16]. Subsequently, the plant material is subjected to surface decontamination, most commonly involving soaking in 70% ethanol (1 min), followed by soaking in sodium hypochlorite (2.5–3.25% NaOCl) for 4–6 min, one more soaking in ethanol (0.5–3 min), and finally rinsing (3–5 times) in sterile distilled water [16,17,18]. Another approach to decontamination involves soaking the plant material in 70% ethyl alcohol for 60 s, flaming it using a burner, and placing it in sterile water [19]. The literature also describes surface disinfection using 2% sodium hypochlorite containing 0.1% Tween 20 [20] or 36% formaldehyde solution for 7 min and a 0.1 M NaOH solution for 10 min [21]. Interestingly, Ding et al. [22] used a different approach for the elimination of microbial contamination of plant specimens. According to an example of their methodology, the stems of *Camptotheca acuminata* were washed with tap water, immersed in 70% ethanol for 1 min, in 0.1% mercuric chloride for 8 min, and then washed six times in sterile distilled water. Subsequently, the final rinsing supernatant was used for endophyte isolation [22]. Mercuric chloride has been well known for its antibacterial properties and application in treating syphilis since the end of the nineteenth century. However, due to its toxicity and severe risk to human health, including renal, gastrointestinal, and central nervous system toxicity, which may result in life-threatening consequences, it is no longer used in medicine. While the risk of inhalation exposure is unsure and not described in the literature, dermal toxicity raises serious concerns [23]. Thus, the use of mercuric chloride for surface decontamination is controversial and should be discouraged.

To ensure proper surface sterilization, different approaches may be used. For example, since the final step in disinfection procedures is rinsing with sterile water, the water used for rinsing can be plated onto tryptic soy agar (TSA), and after incubation, the plates are examined for the presence or absence of microorganism growth [16]. Another approach is to imprint the sterilized tissues onto microbiological media, i.e., nutrient agar (NA) or TSA, and then incubate the plates for any signs of growth [18]. The temperature and time of incubation in the literature vary, with temperatures ranging from 28 to 30 °C and incubation ranging from 7 to 15 days [16,18]. For fungal endophytes, the culturing may be carried out using potato glucose agar (PGA) at room temperature for 14 days [17]. The efficacy of the decontamination procedures can also be tested using control microorganisms. For example, Zinniel et al. [20] sprayed the plant material with a suspension containing the orange-pigmented organism *Clavibacter michiganensis* subsp. *nebraskensis* before the decontamination procedure and subsequently performed colony-counting experiments to evaluate the external bacterial recovery levels. It was shown that external contamination was effectively reduced by more than 10,000-fold [20]. Herein, proper isolation and culturing conditions should also be mentioned. To ensure that endophytes’ isolation and cultivation are carried out without microbial contamination, sterile materials are required under aseptic conditions [19].

The success of surface decontamination procedures determines whether it will be possible to culture the endophytes and prevent the growth of what is supposed to be contamination, namely the epiphytes. This task involves the abovementioned procedures and appears straightforward and easy to conduct, although some issues need careful consideration. As was already described, there are several methods of ensuring the effectiveness of surface decontamination, but the decontamination itself may also affect the endophytic diversity inside the plant material. Therefore, selecting the appropriate sterilant, optimal concentration, and exposure time are of the utmost importance [24,25].

After decontamination, endophytes may be isolated from the plant tissues. This most commonly involves homogenization and culturing steps. The homogenization may be achieved through grinding with water or sterile saline solution (0.9% NaCl) in a sterile mortar [16,18], using mechanical homogenizers [20], or by shaking with sterile metal balls in water for 60 s [19]. Some authors suggest that the tissue extract should be incubated at 28 °C for 3 h to facilitate the release of endophytes from the plant material [16]. Endophytic bacteria are usually isolated using TSA or NA agar [16], whereas for fungi, potato dextrose agar (PDA), PGA, malt extract agar (MEA), Czapek medium, Tryptone Soybean Agar, Tryptone Bovine Extract Agar, or Luria–Bertani medium are used [17,24,26]. Bacterial culturing varies from 5 to 15 days at 28–30 °C [16,18]. Bacterial endophytes have also been shown to grow after 7 days on Columbia agar at room temperature [19]. Fungal endophytes are usually cultured at 25–28 °C for 3 to 20 days, but sometimes several weeks are needed [24]. Afterwards, the characteristic colonies are selected and purified using appropriate media. The distinctive features of pure cultures are evaluated (time of growth), and morphology is described (color, size, and shape) [16,18]. Morphological characteristics of the cultured fungi should be observed on both sides of the colony to describe its size, form, texture, and color, as well as microscopic details of the mycelial septation and the shape and structure of the spores [17,22].

Another approach to microorganism isolation from plant materials is to incubate fragments of surface-decontaminated plant tissues directly on an appropriate growth medium [22,27]. For example, this methodology was used to study endophytic fungi from the leaves and stems of *Zanthoxylum simulans*, wherein leaf and stem segments were placed on 2% MEA, incubated at 25 °C, and observed daily for one month. Growing mycelia were subcultured on MEA and identified. To ensure the selective growth of fungi, the medium was supplemented with penicillin G and streptomycin [27].

Endophytes include microorganisms with versatile metabolic properties, such as those that utilize different substrates in culture media. Hence, it is possible to classify them based on those factors. For example, Ashby’s nitrogen-free medium can be used to evaluate the ability to fix nitrogen, Pikovskaya medium to observe phosphorus solubilization, Aleksandrov medium to determine potassium-resolving ability, or calcium phytate medium to detect phytate-degrading bacteria [28].

Bacterial identification is usually undertaken using a molecular biology-based approach, based on the *rpoB* gene or 16S rRNA gene amplification with subsequent sequencing and alignment with reference sequences retrieved from databases (GenBank database, EzBioCloud, BLAST) [16,18,20,28]. Identification of fungi can be based on morphology and metagenomics by 18S rRNA gene or ITS rDNA (ITS1 or ITS2 internal transcribed spacer) sequencing [17,22,26,27,29]. Additionally, based on the obtained results, it is possible to construct phylogenic trees [18] and a diversity index, which expresses the relative complexity of the endophyte community structure [16,18,29]. Identification procedures may also include culturing methods such as fatty acid and carbon source utilization analyses [20].

To evaluate endophyte metabolites and their biological activity, it may be necessary to obtain large quantities of target microorganisms. Thus, large-scale culturing is advisable, usually using liquid media. For example, Ameen et al. [30] cultured the endophytic fungus *Preussia africana* isolated from *Aloe vera* in a conical flask containing 200 mL of sterilized potato dextrose broth (PDB) with a 28-day incubation period at 28 °C. Subsequently, the crude extract of this fungus was evaluated for biological activities, including antioxidant properties, wound healing, and anticancer activities [30]. Endophytic fungi isolated from *Vitis vinifera* leaves and stems were cultured using 50 mL of pre-sterilized Czapek Dox broth in 250 mL Erlenmeyer flasks at 26 °C and 120 rpm for 7 days [31].

The abovementioned examples indicate that endophytic bacteria and fungi have been isolated from various plants. However, it must be underlined that a significant amount of the symbiotic endophytes are non-cultivable under laboratory conditions [17,24]. Despite this, the study of endophytes that are not prone to laboratory culturing is still possible due to culture-independent DNA-based techniques [24].

## 3. The Bryendophytes

Bryophytes include mosses (*Musci*), liverworts (*Hepaticae*), and hornworts (*Anthrocerotae*). Recently, an increasing interest in the study of bryophytes has been observed, including their biodiversity, phytochemistry, ecological and evolutionary roles, and biotechnological and biomonitoring applications. The bryophytes occupy unique biological niches, which also harbor a substantial diversity of microorganisms, making the study of bryophyte-associated endophytes and epiphytes immensely interesting [32]. The best evidence of a close relationship between bryophytes and endophytes is that the first bryophyte-like land plants, in the early Devonian (400 million years ago), were shown to build endophytic symbioses resembling vesicular–arbuscular mycorrhizas (VAM), even before roots evolved [33]. Mosses often contain endophytic hyphae of VAM fungi, whereas hornworts show a VAM–like symbiosis with glomalean fungi forming arbuscules within their thalli [34,35,36].

A culture-independent approach with PCR-DGGE based on divergent regions of the 16S rRNA gene was performed by Koua et al. [32]. It is worth noting that a limitation of this study was that the plant material was not surface-sterilized, and thus not only the endophytes but also the epiphytes were analyzed. It was found that the microbial community from *Haplocladium microphyllum*, growing on highly-populated soil, included γ-Proteobacteria (*Citrobacter murliniae* and *Klebsiella terrigena*). *Brachythecium buchananii*, growing in the same ecosystem, showed the presence of different γ-Proteobacteria (*Klebsiella intermedia*, *Pseudomonas fluorescens*, *Enterobacter* sp., and *Hafnia* sp.) and some Firmicutes (*Clostridium butyricum* and *C. puniceum*). Whereas, *Trachycystis microphylla* harbored only γ-Proteobacteria, including *Pectobacterium wasabiae*, *Pectobacterium betavasculorum*, *Dickeya dieffenbachiae*, *Serratia proteamaculans*, *Serratia grimesii*, and *Klebsiella oxytoca*. Differences were also found in the microbial flora of bryophytes from virgin rocks: *Brachythecium plumosum* (γ-Proteobacteria: *Salmonella enterica* subsp. *enterica* and Firmicutes: *Anaerobacter polyendosporus*), *Hypnum plumaeforme* (γ-Proteobacteria: *Pseudomonas antarctica*, *Pseudomonas cedrina,* and Firmicutes: *Anaerobacter polyendosporus*, *Clostridium disporicum*, and *Clostridium saccharoperbutylacetonicum*), and *Reboulia hemisphaerica* subsp. *orientalis* (γ-Proteobacteria: *Erwinia rhapontici*, *C. murliniae*, and *Pantoea ananatis*). *Racomitrium japonicum* growing on managed soil showed only the presence of Firmicutes, including *C. saccharoperbutylacetonicum*, and *C. puniceum*. The comparison of identified microorganisms clearly showed differences at the species level regardless of the nature of the ecosystem, indicating a host-dependent microbial community dynamic phenomenon but also showing that selected bacterial genera had a similar distribution among different ecosystems. Noticeably higher bacterial biodiversity was found in the bryophytes collected from highly populated soil and virgin rocks compared to managed soils [37]. Yu et al. [38] studied the endophytic and endolichenic fungal diversity in maritime Antarctica. A total of 93 fungal isolates were obtained from lichens and bryophytes, and most were distributed in six classes of Ascomycota: Dothideomycetes, Eurotiomycetes, Lecanoromycetes, Leotiomycetes, Pezizomycetes, and Sordariomycetes [39].

Examples of endophytes isolated from various bryophytes are summarized in Table 1. Fungal endophytes are commonly isolated from mosses [40,41,42,43,44]. Neslon [45] performed an extensive study of fungal endophytes from the liverwort *Marchantia polymorpha* and observed the presence of at least 45 species belonging to the ascomycete classes Dothideomycetes, Leotiomycetes, Pezizomycetes, Saccharomycetes, and Sordariomycetes. In another study, Nelson and Shaw [46] isolated 86 fungal endophytes from three tissue types: thallus, rhizoids, and gametangiophores of *M. polymorpha* plants were collected from 16 sites in eight US states and one Canadian territory. Additionally, seven endophytes were isolated from fungal fruiting bodies discovered on *M. polymorpha* thalli. The collected plants represented all *M. polymorpha* subspecies, i.e., subsp. *polymorpha*, subsp. *montivagans*, and subsp. *ruderalis*. In total, endophytes were isolated from 67% of plants sampled and mainly belonged to six classes of Ascomycota, including Eurotiomycetes, Pezizomycetes, Saccharomycetes, Leotiomycetes, Dothideomycetes, and Sordariomycetes (the most abundant class). Additionally, two Basidiomycota isolates were obtained, belonging to the Agaricomycetes and Tremellomycetes. Interstingly, rRNA LSU (large subunit) sequencing allowed the identification of non-fungal taxa, including oomycetes, chlorophyte algae, animals such as mites and nematodes, alveolates including *Paramecium* and *Stentor*, and other microbial eukaryotes. This study indicated *Phoma herbarum* as the most common endophyte among the analyzed samples [46]. The effects of fungal isolates on the growth rate of the host organism under laboratory conditions demonstrated high variability from aggressively pathogenic to strongly growth-promoting, but for the majority of endophytes, no detectable changes in the host growth were observed. For example, *Phoma herbarum* isolates did not produce any significant beneficial or detrimental effect on *M. polymorpha* growth [45]. Studies of endophyte biodiversity in *M. polymorpha* may also contribute to the knowledge of ecological factors determining the microbiomes assembly because this species is often found as an early colonizer after fire-induced damage [47,48].

Zhang et al. [57], studied the fungal diversity in three Antarctic bryophyte species: the liverwort *Barbilophozia hatcheri* and the mosses *Chorisodontium aciphyllum* and *Sanionia uncinata*, and found 78 OTUs (Operational Taxonomic Units) from Ascomycota, 13 OTUs from Basidiomycota, 1 OTU from Zygomycota, and 1 OTU from an unknown phylum. The major observed orders were Helotiales, Chaetothyriales, Eurotiales, Sebacinales, and Platygloeales. There were differences at the phylum level among the three bryophyte species, with *Sanionia uncinata* harboring only Ascomycota, *Barbilophozia hatcheri* mostly Basidiomycota, and *Chorisodontium aciphyllum* showing the presence of both Ascomycota and Basidiomycota. Importantly, no OTUs were shared between the three analyzed bryophytes. The moss *Sanionia uncinata* was found to contain more OTUs than the other two bryophyte species, indicating a higher endophyte diversity [58].

The knowledge of bacterial endophytes and bryophytes still needs to be improved. Costa et al. reported that *Nostoc* sp. cyanobacteria were identified in the hornwort *Anthoceros fusiformis* and the liverwort *Blasia pusilla*. Interestingly, some symbiotic *Nostoc* strains were shared by bryophytes growing 2000 m apart [49]. Conversely, Nelson et al. [59], by applying the *rbcL-X* PacBio metabarcoding approach to profile cyanobacterial communities in different hornwort plants, observed that plants growing only a few centimeters apart could have very different sets of cyanobacterial strains. Cyanobacterial endophyte profiles show similarity with those isolated from adjacent soil samples but are highly variable between individual hornworts occupying the same habitat. In this study, no correlations between endophyte communities and distance, time, or host species were observed [59]. The *Nostoc* sp. cyanobacteria were also isolated from the feathermosses *Pleurozium schreberi* and *Hylocomium splendens* [50]. Screening of bacterial endophytes present in the xerophilous moss *Grimmia montana* using a molecular method and cultivated isolates showed that the Proteobacteria and Firmicutes were the dominant phyla, and the most abundant genera included *Acinetobacter*, *Aeromonas*, *Enterobacter*, *Leclercia*, *Microvirga*, *Pseudomonas*, *Rhizobium, Planococcus*, *Paenisporosarcina*, and *Planomicrobium* [56]. Actinomycete bacteria, *Actinomadura physcomitrii*, *Microbispora bryophytorum*, *Actinoallomurus bryophytorum*, and *Streptomyces bryophytorum* were isolated from mosses [51,52,53,54].

The study of endophytic and ectophytic bacterial populations associated with two *Sphagnum* species, *Sphagnum magellanicum* and *Sphagnum fallax*, showed that *Burkholderia* spp. were the dominant group of microorganisms [55]. It was also reported that submerged *Sphagnum* mosses, dominant plants in peat bogs, can utilize methane due to symbiosis with partially endophytic methanotrophic bacteria. This activity of methanotrophic bacteria not only provides the host with a carbon source but also leads to highly effective methane recycling [60].

## 4. Bioactive Natural Products from Bryendophytes

Endophyte metabolites show significant structural diversity and complexity, representing a variety of different chemical classes, such as nitrogenous compounds including alkaloids, peptides, phenolics, polyketides, and terpenoids [13]. Figure 1 shows that within the period 2000–2022, a significant increase in the number of articles concerning compounds produced by endophytes was published. Endophyte metabolites are of increasing interest to researchers, mainly due to their biological properties.

Substances present in plant endophytes can either be produced by the endophytic microorganisms alone or by the plant and the associated endophytes together [10]. The literature on higher plant-associated endophytes is not covered here in any detail other than to show the rapid expansion of this area and the enormous potential that exists to discover new chemistry and biologically active natural products (e.g., [9,37,61,62]). We will instead focus on the area that we believe is less studied and potentially even more fruitful—the bryendophytes.

To the authors best knowledge, there are no reviews on bryendophytes and just several papers on bryophyte metabolites. Figure 2 shows that within, the period 2000–2022, only 19 papers were published.

In this section, we focus on biologically active compounds found in the endophytes associated with the bryophytes: mosses, liverworts, and hornworts. The structures of several bryendophyte metabolites are presented in Figure 3, Figure 4, Figure 5, Figure 6 and Figure 7.

### 4.1. Antibacterial and Antifungal Activity

Research by Guo et al. [33] concerned the chemical components and antifungal and anticancer properties of ether extracts of *Scapania verrucosa* and its endophytic fungus, *Chaetomium fusiforme*. The analysis of the ether extracts from the thalli of *S. verrucosa* and the isolated endophytic fungus showed only a minor correlation in their chemical composition. The major compounds of *S. verrucosa* were sesquiterpenoids of the aromadendrene, aristolene, and calarene types, as well as spathulenol. The presence of phytol, 1-octen-3-ol, and hexadecanoic acid was also confirmed. In the ether extract of the broth of *C. fusiforme,* the following compounds were detected as the major components: 3-methyl-valeric acid methyl ester, butane-2,3-diol, and acetic acid. Interestingly, despite differences in composition, both extracts exerted antifungal activity against *Candida albicans* ATCC76615, *Cryptococcus neoformans* ATCC32609, and *Aspergillus fumigatus* with IC_80_ (80% inhibitory concentration) values between 8 and 64 μg/mL [33].

The antibacterial properties of compounds from liverwort endophytes were studied by Ali et al., [63], who isolated a new prenylated indole alkaloid, *ent*-homocyclopiamine B (**1**), bearing an alicyclic nitro group along with 2-methylbutane-1,2,4-triol from the endophytic fungus *Penicillium concentricum* of the liverwort *Trichocolea tomentella*. The antibacterial activity of *ent*-homocyclopiamine B (**1**) was tested against a panel of Gram-positive and Gram-negative strains. Initial testing using agar plates inoculated with bacteria showed that compound **1** inhibited the growth of *Bacillus subtilis* ATCC 6633 and *Mycobacterium smegmatis* NRRL B-14646. Subsequently, the broth microdilution assay revealed that the growth of *B. subtilis* ATCC 6633, *Rhodococcus jhostii* RHA1, and *Corynebacterium glutamicum* NRRL B-2784 was inhibited by 30% when treated with 100 µM of *ent*-homocyclopiamine B (**1**). This new compound only exhibited slight growth inhibition against selected Gram-positive strains, while all tested Gram-negative bacteria were not susceptible [63].

A crude extract of *Mortierella alpina*, a fungal endophyte isolated from the Antarctic moss *Schistidium antarctici*, demonstrated noteworthy antioxidant activity (EC_50_ of 48.7 μg/mL) and antibacterial activity against *Escherichia coli* (MIC of 26.9 μg/mL), *Pseudomonas aeruginosa*, and *Enterococcus faecalis* (both with an MIC of 107 μg/mL) [42].

### 4.2. Immunosuppressive Properties

Chemical analysis of the ethyl acetate extract of *Aspergillus sydowii*, isolated from the Chinese liverwort *Scapania ciliata*, showed the presence of new xanthone derivatives, namely sidexanthone A (**2**), sidexanthone B (**3**), and 13-*O*-acetylsidovinine B (**4**). Apart from these, the presence of seven other compounds was also confirmed: 1-hydroxy-6,8-dimethoxy-3-methylanthraquinone, 8-hydroxy-6-methyl-9-oxo-9H-xanthene-1-carboxylic acid methyl ester, 1,9-dihydroxy-3-(hydroxymethyl)-10-methoxydibenzo[b,e]oxepin-6,11-dione, pinselin, monilifenone, emodin (**5**), and questin (**6**). To evaluate the immunosuppressive potential of all isolates, the in vitro IC_50_ values against concanavalin A (Con A)-induced and lipopolysaccharide (LPS)-induced proliferation of murine splenic lymphocytes were evaluated using the MTT assay with cyclosporin A as a positive control. Emodin (**5**) and questin (**6**) were found to exert moderate immunosuppressive activity against Con A-induced proliferation with IC_50_ values of 8.45 and 10.10 μg/mL, respectively, and LPS-induced proliferation of 10.25 and 14.10 μg/mL, respectively. Among the new xanthones, only compound (**3**) showed some activity against Con A-induced and LPS-induced proliferation with IC_50_ values of 22.53 and 15.30 μg/mL, respectively [64].

### 4.3. Cytotoxicity and Anticancer Activity

New secondary metabolites have also been isolated from endophytes from the liverwort *Heteroscyphus tener*. From a culture of *Aspergillus fumigatus* associated with this liverwort species, three new compounds, asperfumigatin (**7**), isochaetominine (**8**), and 8′-*O*-methylasteric acid (**9**), were isolated together with nineteen known components. Among the known compounds, chaetominin, brevianamide F, fumitremorgine C, demethoxyfumitremorgine C, 12,13-dihydroxyfumitremorgine C, cyclotryprostatin C, 13-dehydroxycyclotryprostatin C, 20-hydroxycyclotryprostatin B, and spirotryprostatin B can be mentioned. All isolated compounds showed weak anticancer activity in comparison to the ethyl acetate extract of this endophytic fungus, which showed strong cytotoxicity against the PC3 human prostate cancer cell line with an IC_50_ value of 16.72 μg/mL [38].

In a cytotoxic extract from another endophytic fungus, *Aspergillus niger*, also associated with *Heteroscyphus tener*, the presence of naphtho-γ-pyrones was confirmed. These were rubrofusarin-6-*O*-α-D-ribofuranoside (**10**), (*R*)-10-(3-succinimidyl)-TMC-256A1 (**11**), asperpyrone E (**12**), isoaurasperone A, and isoaurasperone F. In addition to the new components, another four known compounds were isolated, namely dianhydroaurasperone C (**13**), aurasperone D (**14**), and asperpyrones A and D. Additionally, the presence of the cyclic pentapeptide malformin A_1_ (**15**) was confirmed. The last compound showed significant in vitro cytotoxicity against the PC3, A2780, H1688, and K562 cell lines, while the crude *Aspergillus niger* extract was characterized by strong cytotoxicity against the HepG-2 cell line. Dianhydroaurasperone C (**13**) was shown to reverse multidrug resistance, and aurasperone D (**14**) caused CNS depression [65].

Another study concerning the liverwort *H. tener* and its endophytic fungus *Aspergillus niger* showed the presence of a new diketopiperazine heterodimer (**16**), asperazine (**17**), cyclo(d-Phe-l-Trp), cyclo(l-Trp-l-Trp), 4-(hydroxymethyl)-5,6-dihydropyran-2-one, valterolactone A, and campyrones A-C. Cytotoxicity tests showed weak activity of asperazine (**17**) (IC_50_ = 56.3 μM) and asperazine A (**16**) (IC_50_ = 56.7 μM) against the A2780 (human ovarian carcinoma) cell line [66].

Studies concerning the ethyl acetate extract of *Marchantia polymorpha* endophytes cultivated in a solid [19] and liquid medium [67] showed the presence of bioactive metabolites belonging to the diketopiperazine class. The most characteristic compounds were cyclo(l-Phe-l-Pro) (**18**), cyclo(l-Leu-l-Pro) (**19**), and their stereoisomers. The cytotoxicity and anticancer potential were assessed using the microculture tetrazolium technique (MTT) towards non-cancerous VERO cells and cancer cells—HeLa, RKO, and FaDu. Results indicate that the crude ethyl acetate extract exerted moderate cytotoxic activity with a significant selectivity, while fractions containing compounds (**18**) and (**19**) showed higher CC_50_ values on cancer cell lines, and a decrease in anticancer selectivity was also observed [67].

A diethyl ether extract of *Chaetomium fusiforme*, an endophytic fungus from *Scapania verrucosa*, showed higher anticancer potential than the extract obtained from the host plant, with IC_50_ values between 9.11 and 31.23 μg/mL, towards LOVO (colorectal cancer), HL-60 (leukemia), and QGY (human papillomavirus-related endocervical adenocarcinoma). Both extracts showed no significant activity on A549 cells [33].

A well-known endophyte in terms of the metabolites it produces is *Penicillium concentricum*, from *Trichocolea tomentella*. Four new compounds were isolated from the culture of this endophytic fungus, including 6-chloro-3,8-dihydroxy-1-methylxanthone, 2-bromogentisinic alcohol (**20**), 6-dehydroxy-6α-bromogabosine C (**21**), and 6-dehydroxy-6β-bromogabosine C (**22**). In addition, the following substances were identified in the extract: epoxydon (**23**) (*R*,*S*)-1-phenyl-1,2-ethanediol, alternariol, epoxide, norlihexanthone, 2-chlorogentisine alcohol, gentisine alcohol, hydroxychlorogentisylquinone, 6-dehydroxy-6α-chlorogabosin C, 6-dehydroxy-6β-chlorogabosin C, as well as griseoxanthone C, griseophenones A and B, griseofulvin, dehydrogriseofulvin, dechlorogriseofulvin, and dehydrodechlorogriseofulvin. Among the isolated compounds, 2-bromogentisyl alcohol (**20**), 6-epimers of 6-dehydroxy-6-bromogabosine C (**21** and **22**), and epoxydon (**23**) displayed modest cytotoxicity to MCF-7 (a hormone-dependent breast cancer cell line) with IC_50_ values of 8.4, 9.7, and 5.7 μM, respectively [68]. Anaya-Eugenio et al. [69] also studied the anticancer potential of compounds isolated from *Penicillium concentricum* against a panel of cancer cell lines, including HeLa (cervical adenocarcinoma), HT-29 (colon cancer), MDA-MB-321, PC-3 (prostate cancer), and DU-145 (prostate cancer). The highest cytotoxic activity against the various cancer cell lines was demonstrated by 2-bromogentisyl alcohol (**20**) and 3-hydroxybenzene methanol (**24**). In the case of epoxydon (**23**), its selective activity (IC_50_ = 1.2 μmol/L) against the DU-145 cell line was proven [69].

Wei et al. [41] isolated a series of metabolites from *Xylaria* sp., a *Hypnum* sp. moss endophytic fungus, including cytochalasin C (**25**), cytochalasin D (**26**), and cytochalasin Q (**27**). Cytochalasins C (**25**) and Q (**27**) showed moderate cytotoxicity against human prostate adenocarcinoma (PC-3M), human non-small cell lung (NCI-H460), human CNS glioma (SF-268), and human metastatic breast adenocarcinoma (MDA-MB-231), as was assessed using a resazurin-based colorimetric (AlamarBlue) assay. However, cytochalasin D (**26**) showed significantly higher anticancer activity towards those cell lines, with CC_50_ values (50% cytotoxic concentration) ranging from 0.22 to 1.03 μM, and additionally inhibiting the human breast (MCF-7) cells (CC_50_ 1.44 μM). Doxorubicin was used as a reference antineoplastic agent, with CC_50_ values ranging from 0.05 to 0.67 μM, depending on the cancer cell line tested [44].

Novel isopimarane diterpenes, smardaesidins A–E (e.g., **28**, **29**), two new 20-*nor*-isopimarane diterpenes, smardaesidins F and G (**30**), as well as sphaeropsidins A (**31**) and C–F, were all isolated from the endophytic fungus *Smardaea* sp. occurring in the moss *Ceratodon purpureus*. A series of sphaeropsidin derivatives were also obtained. Subsequently, all compounds were tested against a panel of cancer cell lines, including human non-small cell lung cancer (NCI-H460), glioma (SF-268), breast cancer (MCF-7), prostate adenocarcinoma (PC-3M), and human metastatic breast adenocarcinoma (MDA-MB-231). It was found that sphaeropsidin A (**31**), sphaeropsidin D (**32**), and 6-*O*-acetylsphaeropsidin A (**33**) showed significant anticancer potential. Notably, sphaeropsidin A (**31**) not only showed activity against all tested cancer cells (1.9–3.0 μM), but also inhibited the migration of MDA-MB-231 cells at sub-cytotoxic concentrations [43].

### 4.4. Allelopathic Activity

An ether extract from a culture of a *Penicillium* sp. from the Chinese liverwort *Riccardia multifida* confirmed the presence of two new metabolites: (3a*R*,9b*R*)-6,9β-dihydroxy-8-methoxy-1-methylcyclopentene[c]isochromene-3,5-dione (**34**) and 6-hydroxyldeoxyfunicone (**35**), together with five known compounds: funicone (**36**), 6-demethylservermistatin (**37**), 3-hydroxymethyl-6,8-dimethoxycoumarin, (+)-mitorubrin, and sequoiamonascin A. The ability of each isolated compound to delay the germination of *Arabidopsis thaliana* seeds was assessed. The most potent activity was shown by 6-hydroxyldeoxyfunicone (**35**) (63.1% at 8 μg/mL). A slightly weaker result of 59.7% was obtained for 6-demethylservermistatin (**37**) [70].

### 4.5. Anti-Inflammatory Activity

Wang et al. isolated two new quinazoline derivatives, versicomides E (**38**) and F (**39**), and ten known compounds from the endophytic fungus *Aspergillus* sp. inhabiting the moss *Trichocoleaceae* sp. All compounds were evaluated for their anti-inflammatory activity with a model to inhibit NO production in lipopolysaccharide (LPS)-activated RAW 264.7 murine macrophages. The compounds notoamide B (**40**), 6-hydroxy-3-methoxyviridicatin (**41**), and 3-*O*-methylviridicatol (**42**) displayed strong inhibitory effects on NO production, with IC_50_ values of 49.85, 22.14, and 46.02 μM, respectively [71].

## 5. Chemical Diversity of Bryendophyte Metabolites

Bioactive metabolites found in bryendophytes show significant structural diversity and complexity. Although there are still few data on the compounds present in bryophyte endophytes, these represent a variety of different chemical classes. As shown in the previous section, bryendophytes produce mainly nitrogen containing compounds, and among them it is worth mentioning indoles (e.g., **1**, **7**), cytochalasin alkaloids (e.g., **25**–**27**), quinazoline derivatives (e.g., **38**, **39**), as well as peptides including diketopiperazines (e.g., **16**–**19**). The presence of cytochalasin alkaloids is noteworthy since these are common yet important metabolites isolated from many genera of ascomycetes and basidiomycetes associated with higher plants. Structurally, cytochalasins have a highly substituted perhydroisoindol-1-one moiety, which is fused with a 9- to 15-membered macrocyclic ring. Many cytochalasins exhibit a wide range of biological activities, such as cytotoxic, antimicrobial, and phytotoxic properties [48,72,73,74]. Indole alkaloids are another group of compounds represented among plant endophytes. The most famous example are endophytes residing in *Catharanthus roseus*, which are capable of synthesizing vinblastine, vincristine, or vincamine, important chemotherapy drugs used to treat numerous cancers [75]. Diketopiperazines are another group of N-containing compounds widespread among endophytic microorganisms, especially Gram-negative bacteria [76].

The fact that alkaloids are among the most characteristic bryendophyte metabolites contradicts the presence of these compounds in the host plant. It is well known that the occurrence of nitrogen-containing compounds in bryophytes is rare in comparison to higher plants. The available data indicated that in the case of liverworts, prenylated indole derivatives have been found in the genus *Riccardia* and isothiocyanates in *Corsinia corriandrina*. Alkaloids have been detected in the moss *Fontinalis squamosa* and in the hornwort *Antoceros agrestis* [77].

Other groups of metabolites identified in bryendophytes include xanthone derivatives (e.g., **2**–**4**), anthraquinones (e.g., **5**, **6**), naphtho-γ-pyrones (e.g., **10**–**14**), phenolic compounds (e.g., **20**–**24**), diterpenoids (e.g., **28**–**33**), and polyketides (e.g., **35**, **36**). All the listed groups of compounds have also been found in endophytes from higher plants.

Isopimarane-type diterpenoids identified in the endophytic fungus *Smardaea* sp. occurring in the moss *Ceratodon purpureus* are the only examples of terpenoids from bryendophytes to date. Terpenoids are a characteristic class of plant metabolites, especially in liverworts. However, endophytes associated with liverwort species produce mainly alkaloids, diketopiperazines, xanthone derivatives, naphtho-γ-pyrones, polyketides, and phenolic compounds. Thus far, no terpenoids have been identified in liverwort endophytes, although the species studied, such as *Scapania verrucosa*, *S. ciliata*, *Heteroscyphus tener*, or *Marchantia polymorpha*, produce diterpenoids of the clerodane type (*Scapania* sp.), labdanoids (*H. tener*), and cuparane, chamigrane, and thujopsene sesquiterpenoids (*M. polymorpha*) as the characteristic compounds. Another two liverwort species, *Riccardia multifida* and *Trichocolea tomentella,* are known to produce bisbibenzyls and isoprenyl phenyl ethers, respectively [3,4,77]. However, the presence of such compounds has not been confirmed in the endophytes of these liverwort species.

The available literature data shows that the bryendophyte compounds described so far are synthesized by the endophytic organisms alone. These natural products correlate with compounds found in endophytes from higher plants. Thus far, there is no evidence that endophytes associated with bryophytes are capable of biosynthesizing compounds that are characteristic of the host plant. To date, no phytochemical correlation has been observed between plant material and their endophytes (Figure 8). Further studies are necessary.

## 6. Conclusions and Future Perspectives

Research on the biology and chemistry of bryophyte endophytes is still very much in its infancy. The examples presented here are mainly from mosses and liverworts; however, even these instances show a profound degree of biological and chemical diversity. What is particularly noteworthy is the ability of bryophyte endophytes, here named for the first time as bryendophytes, to produce multiple classes of natural products that are at the intersection of classical phytochemicals and microbial natural products. This area of overlap is intriguing yet important as it poses research questions on the “transmissibility” of natural products between prokaryotes and eukaryotes and vice versa. The pharmacological activity of bryendophyte metabolites isolated to date is also noteworthy and encouraging of further exploration. Given the niche environments of certain endemic bryophytes, it is highly likely that not only will new bryendophytes be characterized by genetic means, but their chemistry is also likely to elicit a high degree of novelty and biological activity. With this in mind, we will continue our research and focus on novel habitats to collect and study these fascinating organisms.

## Figures and Tables

**Figure 1 molecules-28-03246-f001:**
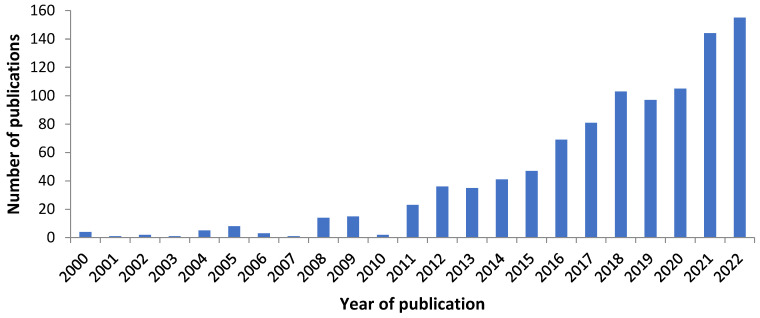
The number of journal articles published between 2000 and 2022 containing the phrase “plant endophyte compounds” and “plant endophyte metabolites” within the title, abstract, or as a keyword (data based on a search from PubMed, 14 January 2023).

**Figure 2 molecules-28-03246-f002:**
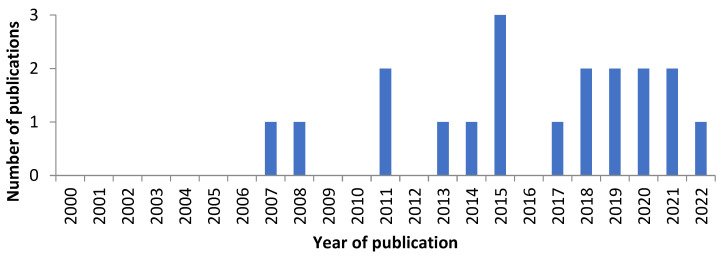
The number of journal articles published between 2000 and 2022 containing the phrase “bryophyte endophyte compounds”, “liverwort endophyte compounds”, “moss endophyte compounds”, and “hornwort endophyte compounds” within the title, abstract, or as a keyword (data based on a search from PubMed, 14 January 2023).

**Figure 3 molecules-28-03246-f003:**
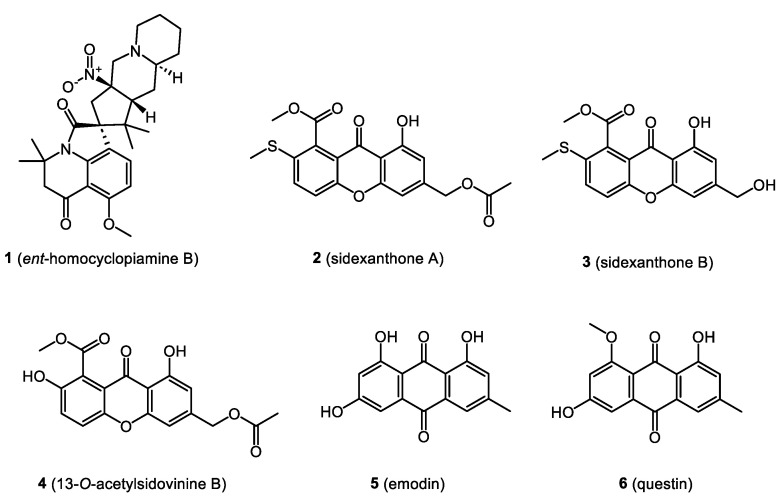
Bryendophyte metabolites with antimicrobial and immunosuppressive properties.

**Figure 4 molecules-28-03246-f004:**
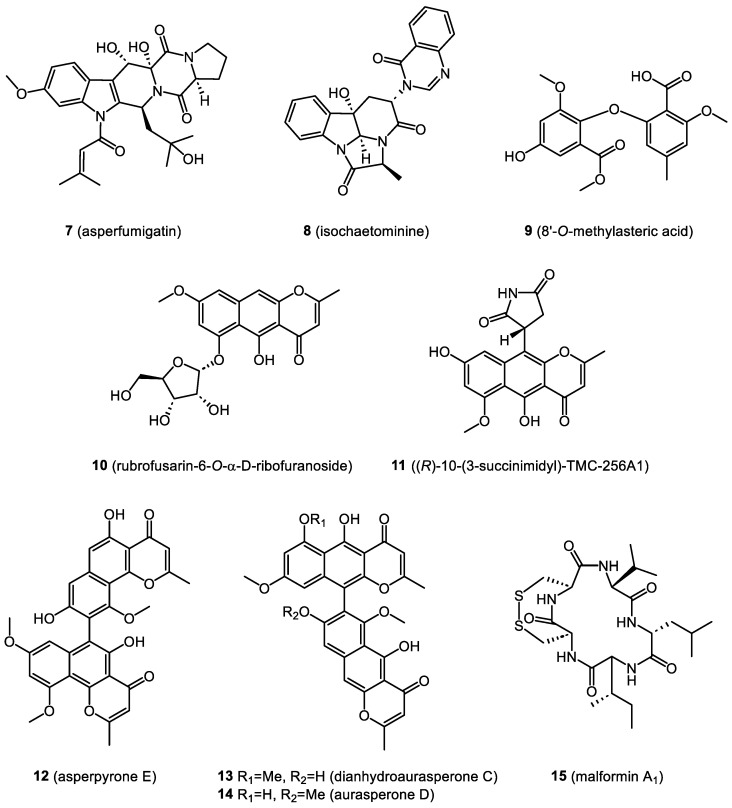
Bryendophyte metabolites (**7**–**15**) with cytotoxic and anticancer activities.

**Figure 5 molecules-28-03246-f005:**
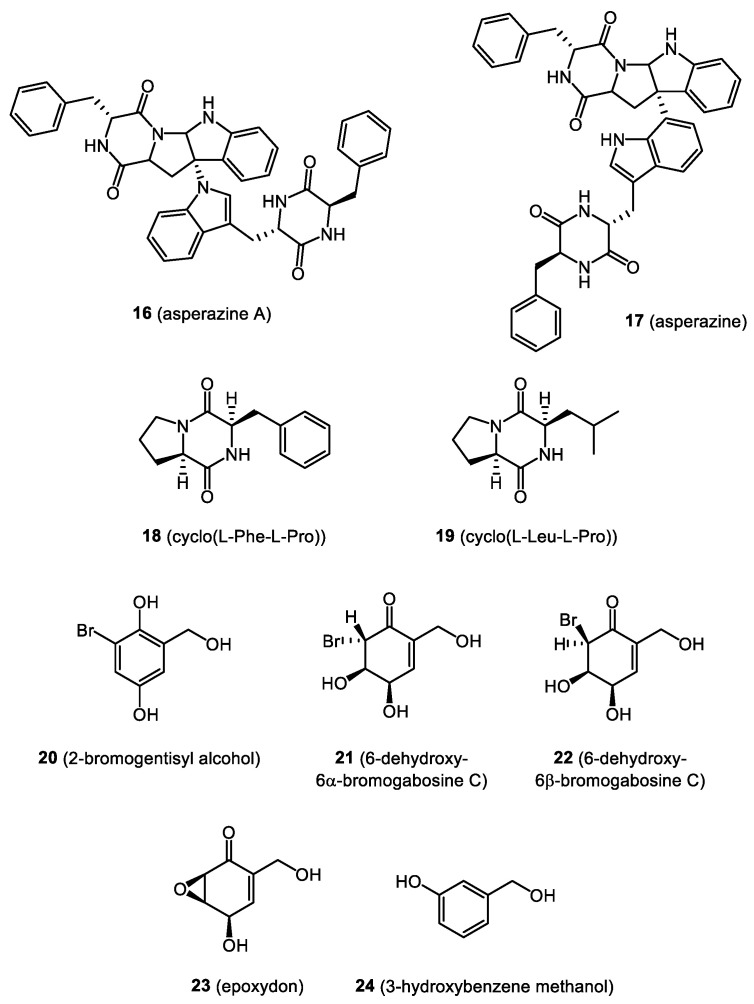
Bryendophyte metabolites (**16**–**24**) with cytotoxic and anticancer activities.

**Figure 6 molecules-28-03246-f006:**
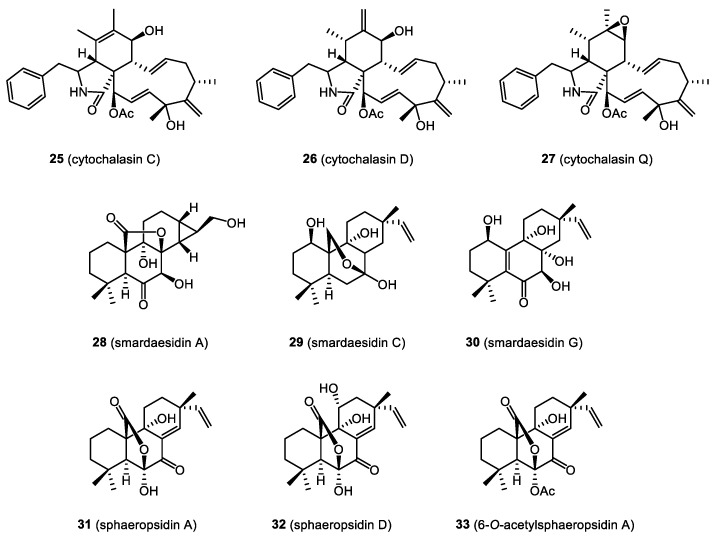
N-containing compounds (**25**–**27**) and pimarane diterpenoids (**28**–**33**) with cytotoxic and anticancer activities.

**Figure 7 molecules-28-03246-f007:**
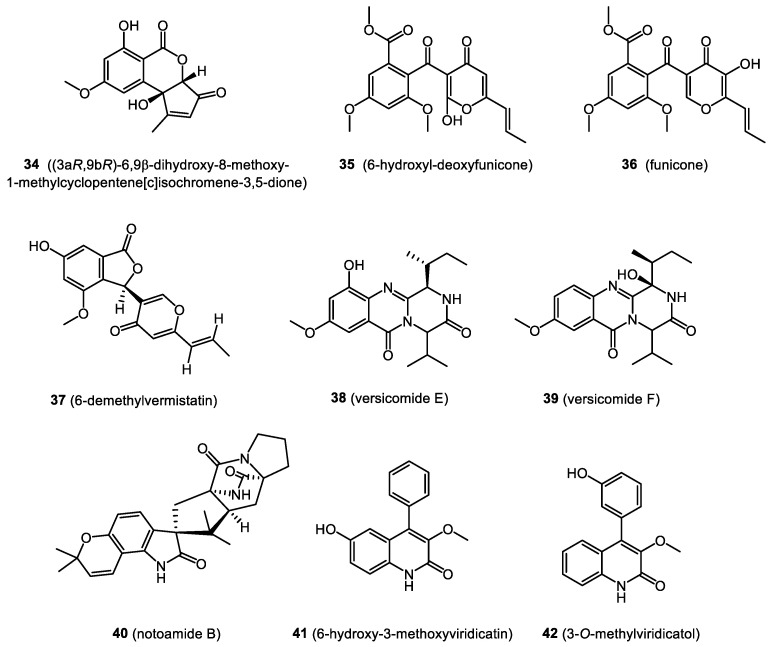
Bryendophyte metabolites (**34**–**42**) with allelopathic and anti-inflammatory activities.

**Figure 8 molecules-28-03246-f008:**
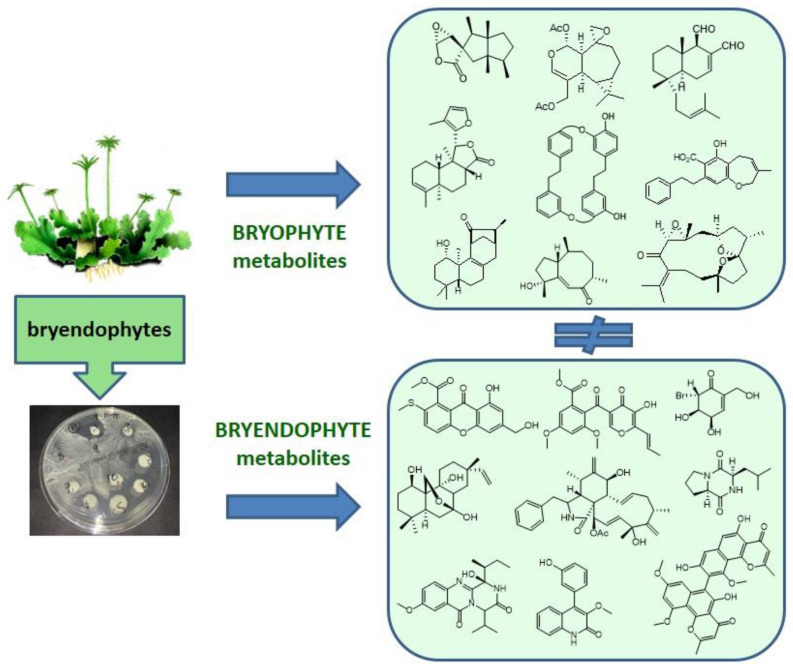
Graphic representation of bryophyte and bryendophyte metabolites.

**Table 1 molecules-28-03246-t001:** Biodiversity of endophytes isolated from selected bryophytes.

Source	Endophyte	Type	Identification	References
Hornwort *Anthoceros fusiformis*	*Nostoc* sp.	Cyanobacteria	tRNA^Leu^ gene	[49]
Liverwort *Blasia pusilla*	*Nostoc* sp.	Cyanobacteria	tRNA^Leu^ gene	
Feathermosses *Pleurozium schreberi* and *Hylocomium splendens* Liverwort *Blasia pusilla*	*Nostoc* sp.	Cyanobacteria	DNA sequencing; PacBio technology	[50]
Moss *Physcomitrium sphaericum*	*Actinomadura physcomitrii* sp. nov.	Bacteria (actinomycete)	16S rRNA	[51]
Moss **	*Microbispora bryophytorum* sp. nov.	Bacteria (actinomycete)	16S rRNA	[52]
Moss **	*Actinoallomurus bryophytorum* sp. nov.	Bacteria (actinomycete)	16S rRNA	[53]
Moss **	*Streptomyces bryophytorum* sp. nov.	Bacteria (actinomycete)	16S rRNA	[54]
Mosses *Sphagnum magellanicum* and *Sphagnum fallax*	*Burkholderia* sp., *Rahnella aquatilis*, *Paenibacillus polymyxa*, *Pseudomonas tolaasii*, *Hafnia* sp., *Microbacterium phyllosphaerae*, and *Streptomyces purpurascens*	Bacteria	16S rRNA	[55]
Moss *Grimmia montana*	Gammaproteobacteria: *Acinetobacter* sp., *Leclercia* sp., *Aeromonas* sp., *Aeromonas* sp., et al.	Bacteria	16S rRNA	[56]
	Alphaproteobacteria: *Rhizobium* sp., *Brevundimonas* sp., *Methylobacterium* sp., et al.			
	Betaproteobacteria: *Bordetella* sp., *Comamonas* sp., *Methylophilus* sp. et al.			
	Firmicutes: *Planococcus* sp., *Planomicrobium* sp., *Bacillus* sp., et al.			
Moss Hylocomiaceae **	*Coniochaeta nivea* sp. nov.	Fungi	ITS rDNA	[40]
Moss *Schistidium antarctici*	*Mortierella alpina*	Fungi	18S rRNA	[42]
Moss *Ceratodon purpureus*	*Smardaea* sp.	Fungi	ITS rDNA LSU rDNA	[43]
Mosses *Polytrichum juniperinum*, *Aulacomnium palustre*, and *Sphagnum fuscum*	*Cladophialophora minutissima*	Fungi	ITS 18S SSU (18S rRNA)	[41]
Moss, *Hypnum* sp. (Hypnaceae)	*Xylaria* sp. *	Fungi	ITS rDNA, LSU rDNA	[44]
Liverwort, *Marchantia polymorpha*	*Biscogniauxia mediterranea*	Fungi	ITS1, ITS2, LSU rDNA	[45]
	*Colletotrichum truncatum*			
	*Daldinia loculata*			
	*Hypoxylon submonticulosum*			
	*Nemania serpens*			
	*Phoma herbarum*			
	*Toxicocladosporium irritans*			
	*Xylaria cubensis*, *Xylaria arbuscula*			
	*Candida* sp., *Coniochaetaceae* sp., *Helotiaceae* sp., *Pezizales* sp., *Plectosphaerella* sp., *Pleosporales* sp., *Schizothecium* sp. *			

*—awaiting further studies and identification; LSU rDNA (nuclear ribosomal large subunit); ITS rDNA (ITS1 or ITS2 internal transcribed spacer). **—details were not provided.

## Data Availability

The data presented in this study are available upon request from the corresponding authors.

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
