# Peer review of "Bioactive Compounds Produced by Endophytic Microorganisms Associated with Bryophytes—The “Bryendophytes”"

_molecules, 2023, doi:10.3390/molecules28073246_

Round 1

Reviewer 1 Report

This review manuscript considers the subject of bioactive compounds produced by endophytic microorganisms associated to the Bryophyta which, as the authors point out, has been neglected or scarcely considered so far; yet, it is likely to disclose novel findings in terms of species occurrence and chemodiversity of these symbionts. The paper is quite well written, but in my opinion it requires major modification in order to be considered for publication in Molecules, as per the below list of corrections/adjustments.

Abstract

It is quite long, and making reference to the human microbiome seems too remote. I propose to cut text at lines 12-18, and to start as follows: ‘Mutualistic interactions between plants and endophytic microorganisms are diverse and….’

Introduction

Lines 53-59: there is some redundancy in this text. A more essential form could be: ‘This review highlights the importance of studying endophytic microorganisms associated with plants, especially with bryophytes. For the first time we describe this fascinating class of microbes as the bryendophytes, to capture the essence of their ecological association. Particularly, we focus on their isolation, cultivation, and the chemical diversity of their bioactive compounds, by accessing the pertinent information through the PubMed, Scopus, Web of Science and Google Scholar databases.’

Section 2

Line 61-72: this part is repetitive of concepts already presented in the introduction. Authors should integrate it in the previous section along with the cited references, and directly start this section with the subsection 2.1;

line 79: correct to ‘root epidermal’;

line 89: delete ‘then’;

line 92: remove comma between ‘sterile’ and ‘distilled’;

lines 97-108: I do not see the point in describing a method based on a dangerous sterilizing agent, which use should be discouraged;

lines 149-152: this sentence requires some adjustment, i.e. ‘Morphological characteristics of the cultured fungi should be observed on both sides of the colony, so to describe its size, form, texture and colour, and microscopic details of the mycelial septation and of the shape and structure of the spores’;

line 159: correct ‘the MEA’ to ‘the medium’;

Line 180: ‘Preussia africana’ not in parentheses;

line 184: delete ‘cultures’;

line 196: correct ‘present in’ to ‘produced by’;

Section 2.2

Since authors decided to focus this review on compounds produced by the ‘bryendophytes’, it is not acceptable that they dedicate a large part of the manuscript (about 7 pages) to the biosynthetic capacities of endophytes of other plants. This section should only offer a general outlook on the latter aspect, also by making reference to some of the many fine reviews which have been published in the field in recent years; in this respect, figure 1 is ok, while the detailed reference to specific strains/products, herein provided along with chemical structures, is definitely not pertinent should be removed.

Section 3

I suggest to change title in ‘The Bryendophytes’, since the term has been previously defined;

line 426: delete ‘are the first terrestrial plants and’, since this information has been previously provided;

line 454: correct ‘sub.’ to ‘subsp.’, and delete ‘Citrobacter murliniae, Pantoea ananatis,’ (both annotated twice);

lines 456-457: use the abbreviated names C. saccharoperbutylacetonicum and C. puniceum, as these species have been previously mentioned in the text. This aspect should be carefully checked also for other species names (including plant names) throughout the manuscript;

line 468: correct to ‘Nelson’;

line 471: correct ‘orders’ to ‘classes’;

Table 1

Reduce font size to 9 and adapt column width for a better alignment; remove species authority at the 4th line.

Line 501: change to ‘Operational Taxonomic Units (OTUs)’;

line 531: correct to ‘spp.’, as the subject is meant to be plural; otherwise, correct to ‘showed Burkholderia sp. to be the…’;

Section 4

Number subsections (4.1, 4.2, etc.);

line 540: use either ‘bryophyte endophytes’ or ‘bryendophytes’;

line 564 and line 651: use ‘C. fusiforme’;

lines 569-570: reduce to ‘Ali et al. [90] isolated…’;

line 582: Lycopodium clavatum belongs to the Lycopodiopsida, which are not a subject in this review; hence, this part of the text and ref. [91] must be removed;

line 609: correct to ‘the new’;

lines 635-636: correct to ‘the crude extract’;

line 692-704: despite the trivial name, Tillandsia usneoides is a flowering plant rather than a moss; hence, this part of the text and ref. [98] must be removed;

line 734: ‘from’ in lowercase initial;

line 735: ‘Trichocoleaceae’ not in italics;

line 758: correct to ‘focus on’.

Reviewer 2 Report

The subject of the review is consistent with the scope of the Journal. The present review is focused on issues about bioactive compounds produced by endophytic microorganisms associated with bryophytes – the “Bryendophytes.

The authors showed both information on bioactive compounds produced by endophytes isolated from higher plants and those produced by endophytes liberated from bryophytes. Such summaries are available and this one does not add much new information. In my opinion, the presentation of information on the production of bioactive compounds by micro-organisms as associated with higher plants is unnecessary. The authors should focus only on presenting information concerning the topic of the review. Then this review will be valuable and will present a new and uncommon review.

Therefore, I think that article can be published in scientific Molecules after some changes (major revision).

-        -  please remove the information on bioactive compounds produced by microorganisms associated with higher plants in favour of providing deeper, more accurate information on compounds produced by bryendophytes.

-         - please insert a graphic representation of this summary, this will help to attract the reader but also reinforce the understanding of the topic.

Round 2

Reviewer 1 Report

Authors improved the manuscript following previous indications. Hence, in my opinion the manuscript could be accepted for publication after minor revision concerning the following adjustments: line 150: change 'of' to 'for'; line 161: use abbreviation 'TSA' for 'trypton soybean agar', as it was previously mentioned at line 130; line 162: 'Tryptone Bovine Extract Agar' in lowercase initials; line 194: correct to 'phylogenetic'; line 249: correct to 'bacterial diversity'; Table 1: check missing reference at the second line; in case it is the same as the first line, then separation between the lines should be removed. Also, blank line before 'Anthoceros fusiformis' must be removed; lines 289-290: change to '...78 operational taxonomic units (OTUs) from Ascomycota, 13 OTUs from...'; line 291: change 'major' to 'most'; lines 329-330: a more correct form of this sentence could be '...-2022, the number of articles concerning compounds produced by endophytes has significantly increased.'; line 347: correct to '...authors' best...'; line 427: correct to 'chaetominine'; line 455: add spaces before measure units; line 457: use the abbreviated form 'M. polymorpha'; an accurate check concerning the use of the abbreviated form after the first mention of full names of plant and fungal species should be done throughout the manuscript; lines 545 and 549: correct to 'demethylvermistatin'.

Author Response

Reviewer

Authors improved the manuscript following previous indications. Hence, in my opinion the manuscript could be accepted for publication after minor revision concerning the following adjustments:

line 150: change 'of' to 'for';

line 161: use abbreviation 'TSA' for 'trypton soybean agar', as it was previously mentioned at line 130;

line 162: 'Tryptone Bovine Extract Agar' in lowercase initials;

line 194: correct to 'phylogenetic';

line 249: correct to 'bacterial diversity';

Table 1: check missing reference at the second line; in case it is the same as the first line, then separation between the lines should be removed. Also, blank line before 'Anthoceros fusiformis' must be removed;

lines 289-290: change to '...78 operational taxonomic units (OTUs) from Ascomycota, 13 OTUs from...';

line 291: change 'major' to 'most';

lines 329-330: a more correct form of this sentence could be '...-2022, the number of articles concerning compounds produced by endophytes has significantly increased.';

line 347: correct to '...authors' best...';

line 427: correct to 'chaetominine';

line 455: add spaces before measure units;

line 457: use the abbreviated form 'M. polymorpha'; an accurate check concerning the use of the abbreviated form after the first mention of full names of plant and fungal species should be done throughout the manuscript;

lines 545 and 549: correct to 'demethylvermistatin'.

Answer: Thank you very much for all your comments and corrections to the language. We appreciate it very much. All corrections are included in revised manuscript.

Reviewer 2 Report

Many thanks to the Authors for improving the manuscript. There is definitely a better perception of it now.  I  think that the manuscript can be accepted for printing.

Author Response

Reviewer

Many thanks to the Authors for improving the manuscript. There is definitely a better perception of it now.  I think that the manuscript can be accepted for printing.

Answer: Thank you very much for your comment. We want to extend our appreciation for taking the time and effort necessary to provide guidance to improve our paper